# Explaining How a Neural Network Play the Go Game and Let People Learn

## Abstract

The AI model has surpassed human players in the game of Go (Granter et al., 2017; Fang et al., 2018; Intelligence, 2016), and it is widely believed that the AI model has encoded new knowledge about the Go game beyond human players. In this way, explaining the knowledge encoded by the AI model and using it to teach human players represent a promising-yet-challenging issue in explainable AI. To this end, mathematical supports are required to ensure that human players can learn accurate and verifiable knowledge, rather than specious intuitive analysis. Thus, in this paper, we extract interaction primitives between stones encoded by the value network for the Go game, so as to enable people to learn from the value network. Experiments show the effectiveness of our method.

## 1 Introduction

The explanation for AI models has gained increasing attention in recent years. However, in this paper, we consider a new problem, *i.e.*, if an AI model has achieved superior performance in a task to human beings, then how can we use the explanation of this AI model to provide new insights and teach people to better conduct the task? In this study, we focus on AI models designed for the game of Go. It is because AI models for the Go game are regarded to have surpassed human players, and may learn the inference logic beyond current human understandings of the game of Go (Granter et al., 2017; Fang et al., 2018; Intelligence, 2016). Therefore, we aim to explain the complex inference logic encoded by these AI models to teach human players to play the Go game.

The current AI model usually jointly uses the value network, policy network, and Monte Carlo tree search to play the Go game. To simplify the explanation, in this study, we only explain shape patterns[1] encoded by the value network. However, the elaborate strategies for the Go game proposes high requirements for the trustworthiness of the explanation. In particular, the Go game is widely considered as much more complex than most other games (Shin et al., 2021; 2020). In the Go game, even minor alterations to 1-2 stones on the board can fundamentally change the result of the game. Therefore, the explained insights into the value network are supposed to be proved by theories, be verified by experiments, and be accountable for errors, instead of specious intuitive analysis.

Specifically, the explanation method needs to address following two new challenges to provide a provable and verifiable explanation for the inference logic of the value network. (1) For models for most other applications, we can explain the model by simply visualizing implicit appearance patterns encoded by the model (Simonyan et al., 2014; Dosovitskiy & Brox, 2016; Yosinski et al., 2015; Zeiler & Fergus, 2014), or estimating attributions of different input variables (Lundberg & Lee, 2017; Selvaraju et al., 2017; Zhou et al., 2016; Zintgraf et al., 2017). However, due to the high complexity of the Go game, we need to explain explicit primitive shape patterns[1], which are used by the neural network as primitive logic to play the game. (2) The rigor of the explanation of shape patterns[1] must be guaranteed in mathematics. It is because the superior complexity of the Go game can easily lead to specious or groundless explanations that will misguide human players.

To this end, (Li & Zhang, 2023; Ren et al., 2023b) have attempted to define and extract interactions encoded by a DNN. Let us consider the Go game shown in Figure 1. Given a game state $x$ with $n$ stones on the board, $N = \{1, 2, ..., n\}$, we use interactions to explain the advantage score of white stones estimated by the value network. The value network may encode the interaction between

---

[1]Shape patterns, refer to the various shapes formed by the arrangement of stones on the board.

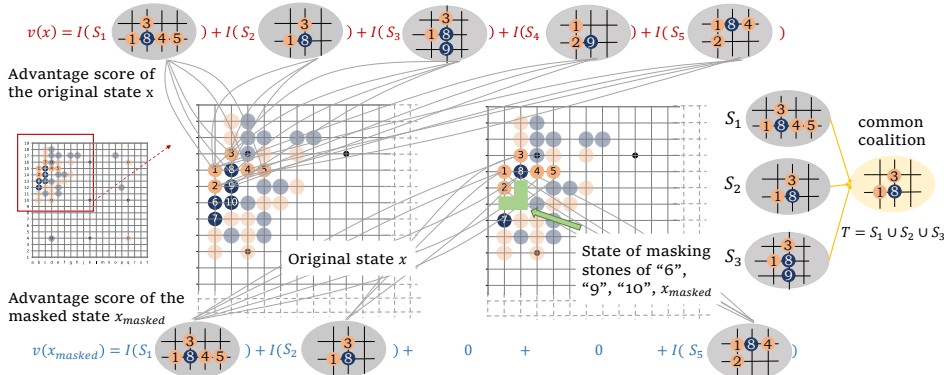

Figure 1: Interactions encoded by the value network. Each interaction $S$ represents a specific shape pattern corresponding to an AND relationship among a set of stones in $S$. The stones $x_6, x_9, x_{10}$ are removed in the masked board state $\boldsymbol{x}_{\text{masked}}$. Because the stone $x_9$ is removed from the board, the interactions $S_3$ and $S_4$ are deactivated in the masked board state $\boldsymbol{x}_{\text{masked}}$, i.e., $I(S_3|\boldsymbol{x}_{\text{masked}}) = 0$, and $I(S_4|\boldsymbol{x}_{\text{masked}}) = 0$. The coalition $T = \{1, 3, 8\}$ participates in different interactions $S_1, S_2, S_3$.

stones in $S_1 = \{1, 3, 4, 5, 8\}$. Each interaction $S \subseteq N$ represents a specific shape corresponding to an AND relationship between stones in $S$. When all stones in $S$ are present on the board, the interaction $S$ is activated and make an effect $I(S)$ on the output of the value network. The removal of any stone in $S$ will deactivate the interaction effect from the network output.

However, we find that the previous interaction-based explanation cannot be directly used to discover novel shapes from the value network. We overcome following three major challenges.

● Besides explaining AND relationships between stones, we need to extend the original definition of interactions to further explain OR relationships between stones encoded by the value network. *I.e.*, the presence of any stones in a set of positions $S$ would make a certain effect $I_{\text{or}}(S)$.

● We find that the advantage score estimated the value network is usually shifted/biased, when white stones are far less or far more than black stones in the board. To this end, we develop a method to alleviate the shifting problem to simplify the explanation.

● We need to show that given a certain state $\boldsymbol{x}$ on the Go board, the outputs of the value network can always be mimicked by a small number of AND interactions and OR interactions, no matter how we randomly remove stones from the board.

Furthermore, we notice that shape patterns for the Go game are usually quite complex, *i.e.*, each interaction often contains a large number of stones. Too complex shape patterns are usually considered as the specific shapes memorized by the value network for a specific state, instead of a common shape patterns shared by different states. Thus, the complexity of shape patterns boosts the difficulty of understanding the Go game. Thus, we further identify some common combinations of stones that are shared by different interactions/shapes, namely *coalitions*. For example, in Figure 1, the interactions $S_1 = \{1, 3, 4, 5, 8\}, S_2 = \{1, 3, 8\}, S_3 = \{1, 3, 8, 9\}$ all contain the coalition T={1, 3, 8}. We apply (Xinhao Zheng, 2023) to estimate the attribution of each coalition to help human players understand the DNN's logic. We collaborate with professional human Go players to compare the interactions or coalitions encoded by the value network with the human understanding of the Go game, so as to discover advanced shapes beyond human understanding.

We conducted experiments to evaluate attributions of some manually-annotated coalitions, and co-operated with professional human Go players to further explain these attributions. We found many cases that fitted to human understandings of shape patterns, as well as a few cases that conflicted with normal understandings of shape patterns, which provided new insights into the Go game.

## 2 RELATED WORK

Many methods have been proposed to visualize the feature/patterns encoded by the DNN (Simonyan et al., 2014; Dosovitskiy & Brox, 2016; Yosinski et al., 2015; Zeiler & Fergus, 2014), or to estimate

the attribution/importance of each input variable (Lundberg & Lee, 2017; Selvaraju et al., 2017; Zhou et al., 2016; Zintgraf et al., 2017).

However, the demand to teach human players proposes higher requirements for the explanation method. We need to clarify the explicit logic used by the DNN, which is supposed to be theoretically guaranteed and experimentally verified, instead of a specious understanding. To this end, (1) Ren et al. (2023a) and Ren et al. (2023b) have proven that a well-trained DNN usually encodes a small number of interactions, and the output score of the DNN on a certain input sample can always be well mimicked by numerical effects of a few salient interactions, no matter how the input sample is randomly masked. (2) Li & Zhang (2023) have further found the considerable transferability of interactions over different samples and over different DNNs. (3) Interaction primitives (the Harsanyi interaction) can explain the elementary mechanism of previous explanation metrics, *e.g.*, the Shapley value (Shapley, 2016), the Shapley interaction index (Grabisch & Roubens, 1999), and the Shapley Taylor interaction index (Sundararajan et al., 2020a).

Despite of above findings, explaining the DNN for the Go game still proposes new challenges. To this end, we extend the AND interaction to the OR interaction, solve the saturation problem of the advantage score, and compute attributions of common coalitions shared by different interactions, thereby obtaining concise and accurate explanation for shape patterns in the value network.

## 3 EXPLAINING THE INFERENCE LOGIC OF THE VALUE NETWORK

### 3.1 PRELIMINARIES: INTERACTIONS ENCODED BY THE DNN

• **Definitions of the interaction.** In this paper, we use the value network $v$ for the game of Go as an example to introduce interactions between different stones encoded by the value network. The value network uses the current state $x$ on the board to estimate the probability $p_{\text{white}}(x)$ of white stones winning. To simplify the notation, let us use $x = \{x_1, x_2, ..., x_n\}$ to denote both positions and colors of $n$ stones in the current state. We consider these $n$ stones, including both white and black stones, as input variables[2] of the value network, which are indexed by $N = \{1, 2, ..., n\}$. We set a scalar $v(x) = \log(\frac{p_{\text{white}}(x)}{1 - p_{\text{white}}(x)}) \in \mathbb{R}$ as the advantage of white stones in the game.

In this way, Harsanyi (1963) has proposed a metric $I(S)$, namely the *Harsanyi dividend* or the *Harsanyi interaction*, to measure the interaction between each specific set $S \subseteq N$ of input variables (stones) encoded by the model $v$. Each interaction $S$, *e.g.*, $S_2 = \{1, 3, 8\}$ in Figure 1, denotes a certain shape of stones. If all stones in $S$ are placed on the board, then the interaction will make a numerical effect on the advantage score $v(x)$. Thus, we can consider the interaction as an AND relationship $I(S_2) = w_{S_2} \cdot [exist(x_1) \& exist(x_3) \& exist(x_8)]$ encoded by $v$, where the Boolean function $exist(\cdot) = 1$ when the stone is placed on the board; $exist(\cdot) = 0$ when the stone is removed. Otherwise, the removal of any stones in $S$ will deactivate the effect, *i.e.*, making $I(S|x_{\text{masked}}) = 0$. Such an interaction effect can be measured from the value network based on the following definition.

$$I(S) \overset{\text{def}}{=} \sum_{T \subseteq S} (-1)^{|S| - |T|} \cdot v(x_T) \tag{1}$$

where $x_T$ denotes the state when we keep stones in the set $T$ on the board, and remove all other stones in $N \setminus T$. Thus, $v(x_T)$ measures the advantage score of the masked board state $x_T$.

• **Sparsity of interactions and interaction primitives.** Although we can sample $2^n$ different subsets of variables from $N$, *i.e.*, $S_1, S_2, S_3, ..., S_{2^n} \subseteq N$, Li & Zhang (2023); Ren et al. (2023b) have discovered and proved that a well-trained DNN usually only encodes a small number of interactions in some common conditions[3]. In other words, most interactions $S \subseteq N$ defined in Equation (1) usually have almost zero effect, $I(S) \approx 0$. Only a few interactions $S \in \Omega_{\text{salient}}$ have considerable effects, *s.t.* $|\Omega_{\text{salient}}| \ll 2^n$. In this paper, we set a threshold $\xi = 0.15 \cdot \max_S |I(S)|$ to select salient interactions, $\Omega_{\text{salient}} = \{S : |I(S)| > \xi\}$.

We can consider the small number of salient interactions $S \in \Omega_{\text{salient}}$ as primitive inference patterns encoded by the value network, namely **interaction primitives**, because Theorem 1 shows that these

---

[2]Although the actual input of the value network is a tensor (Silver et al., 2016), in this paper, we use $x = \{x_1, x_2, ..., x_n\}$ to denote the input of the value network for simplicity.

[3]Please see Appendix B for more detailed introductions of common conditions.

interaction primitives can always well mimic the network outputs no matter how we randomly mask the input sample $\boldsymbol{x}$.

**Theorem 1** (proved by Ren et al. (2023a)). *Let us randomly mask a given input sample $\boldsymbol{x}$ to obtain a masked sample $\boldsymbol{x}_T$. The output score of the DNN on all $2^n$ randomly masked samples $\boldsymbol{x}_T$ w.r.t. $\forall T \subseteq N$ can all be approximated by the sum of effects of a small number of salient interactions.*

$$v(\boldsymbol{x}_T) = \sum_{S \subseteq T} I(S) \approx \sum_{S \in \Omega_{salient}, S \subseteq T} I(S) \tag{2}$$

Theorem 1 shows that when we remove stones in a random set $N \setminus T$ from the board state $\boldsymbol{x}$ and obtain a masked state $\boldsymbol{x}_T$, the output score $v(\boldsymbol{x}_T)$ of the value network on $\boldsymbol{x}_T$ can be explained by a small number of salient shape patterns encoded by the value network.

• **Complexity of the interaction primitive.** The complexity of an interaction primitive $S$ is defined as the order of the interaction, *i.e.*, the number of stones in $S$, $\text{order}(S) = |S|$. An interaction primitive of a higher order represents a more complex interaction with more stones.

## 3.2 EXTRACTING SPARSE AND SIMPLE INTERACTION PRIMITIVES

To teach people about new patterns to play the game of Go, we first extract interaction primitives encoded by the value network. We consider them as shape patterns used by the value network to estimate the winning probability. To this end, we need to address the following three challenges.

• **Challenge 1. Extending AND interactions to OR interactions.** The original Harsanyi interaction just represents the AND relationship between a set of stones encoded by the network. However, compared to most other applications, the game of Go usually applies much more complex logic (Shin et al., 2021; 2020), so we extend AND interactions in Equation (1) to OR interactions. We simultaneously use these two types of interactions to explain the Go game.

Logically, an OR relationship can be represented as the combination of binary logical operations "AND" and "NOT." For example, we represent the effect of an AND interaction $S = \{1, 2, 3\}$ as $I_{\text{and}}(S) = w_S^{\text{and}} \cdot [exist(x_1)\&exist(x_2)\&exist(x_3)]$, where $\&$ represents the binary logical operation "AND." In comparison, the effect of an OR interaction $S = \{1, 2, 3\}$ is represented as $I_{\text{or}}(S) = w_S^{\text{or}} \cdot [exist(x_1)|exist(x_2)|exist(x_3)] = w_S^{\text{or}} \cdot \{\neg[(\neg exist(x_1))\&(\neg exist(x_2))\&(\neg exist(x_3))]\}$, where $|$ represents the binary logical operation "OR," and the Boolean function $\neg exist(\cdot) = 1$ when the stone is removed, $\neg exist(\cdot) = 0$ when the stone is placed on the board.

Therefore, we consider the advantage score of the value network $v(\boldsymbol{x}_T)$ *w.r.t.* any masked sample, $\forall T \subseteq N$, intrinsically contains the following two terms.

$$\forall T \subseteq N, v(\boldsymbol{x}_T) = v_{\text{and}}(\boldsymbol{x}_T) + v_{\text{or}}(\boldsymbol{x}_T) \tag{3}$$

where the advantage term $v_{\text{and}}(\boldsymbol{x}_T)$ is exclusively determined by AND interactions, and the advantage term $v_{\text{or}}(\boldsymbol{x}_T)$ is exclusively determined by OR interactions. Later, in Equation (6), we will introduce how to automatically learn/disentangle $v_{\text{and}}(\boldsymbol{x}_T)$ and $v_{\text{or}}(\boldsymbol{x}_T)$ from $v(\boldsymbol{x}_T)$.

Just like in Equation (1), the AND interaction is redefined on the advantage term $v_{\text{and}}(\boldsymbol{x}_T)$, and is given as $I_{\text{and}}(S) = \sum_{T \subseteq S}(-1)^{|S|-|T|} \cdot v_{\text{and}}(\boldsymbol{x}_T)$. Similarly, an OR interaction $S$ is measured to reflect the strength of an OR relationship between stones in the set $S$ encoded by the model $v_{\text{or}}$. If any stone in $S$ appears on the board, then the OR interaction $S$ makes an effect $I_{\text{or}}(S)$ on the score $v_{\text{or}}(\boldsymbol{x})$. Only all stones in $S$ are removed from the board, the effect $I_{\text{or}}(S)$ is removed. Thus, we can define the effect $I_{\text{or}}(S)$ of an OR interaction $S$ on the model output $v_{\text{or}}(\boldsymbol{x})$, as follows.[4]

$$I_{\text{or}}(S) = -\sum_{T \subseteq S}(-1)^{|S|-|T|}v_{\text{or}}(\boldsymbol{x}_{N \setminus T}), \quad S \neq \emptyset \tag{4}$$

**Theorem 2** (proved in Appendix C). *The OR interaction effect between a set $S$ of stones, $I_{or}(S)$ based on $v(\boldsymbol{x}_T)$, can be computed as a specific AND interaction effect $I'_{and}(S)$ based on the dual function $v'(\boldsymbol{x}_T)$. For $v'(\boldsymbol{x}_T)$, original present stones in $T$ (based on $v(\boldsymbol{x}_T)$) are considered as being removed, and original removed stones in $N \setminus T$ (based on $v(\boldsymbol{x}_T)$) are considered as being present.*

---

[4] Please see Appendix D for the proof.

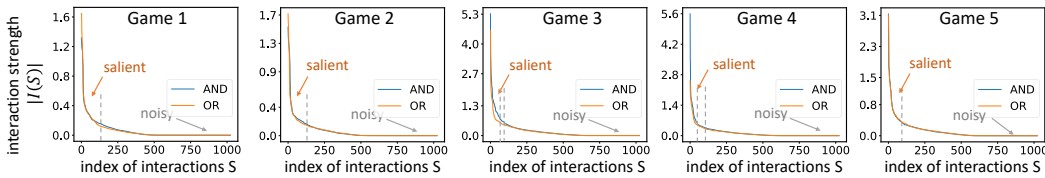

Figure 2: Strength of effects of all AND interactions and OR interactions in descending order. Only a small number of interactions have salient effects on the output of the value network.

**Theorem 3** (proved in Appendix E). *Let the input sample $\boldsymbol{x}$ be randomly masked. There are $2^n$ possible masked samples $\{\boldsymbol{x}_T\}$ w.r.t. $2^n$ subsets $T \subseteq N$. The output score on any masked sample $\boldsymbol{x}_T$ can be represented as the sum of effects of both AND interactions and OR interactions.*

$$v(\boldsymbol{x}_T) = v(\boldsymbol{x}_\emptyset) + v_{and}(\boldsymbol{x}_T) + v_{or}(\boldsymbol{x}_T) = v(\boldsymbol{x}_\emptyset) + \sum\nolimits_{S \subseteq T, S \neq \emptyset} I_{and}(S) + \sum\nolimits_{S \cap T \neq \emptyset, S \neq \emptyset} I_{or}(S) \quad (5)$$

To automatically learn the disentanglement of AND interactions and OR interactions, we set $\forall T \subseteq N, v_{\text{and}}(\boldsymbol{x}_T) = \frac{1}{2}v(\boldsymbol{x}_T) + p_T$ and $v_{\text{or}}(\boldsymbol{x}_T) = \frac{1}{2}v(\boldsymbol{x}_T) - p_T$, which satisfies $\forall T \subseteq N, v(\boldsymbol{x}_T) = v_{\text{and}}(\boldsymbol{x}_T) + v_{\text{or}}(\boldsymbol{x}_T)$, so that the learning of the disentanglement of $v_{\text{and}}(\boldsymbol{x}_T)$ and $v_{\text{or}}(\boldsymbol{x}_T)$ is equivalent to the learning of $\{p_T\}_{T \subseteq N}$. $p_T \in \mathbb{R}$ denotes a learnable bias term. Furthermore, we notice that small unexplainable noises in the network output can be enlarged in interactions[5]. To overcome this problem, we slightly revise the original network output $v(\boldsymbol{x}_T)$ as $v(\boldsymbol{x}_T) + q_T$, where $q_T \in \mathbb{R}$ is a small scalar contained within a small range, $|q_T| < \tau$[6]. The parameter $q_T$ is learned to represent the unavoidable noises in the network output, which cannot be reasonably explained by AND interactions or OR interactions. According to the Occam's Razor, we use the following loss function to learn the sparse decomposition of AND interactions and OR interactions.

$$\min_{\boldsymbol{p}, \boldsymbol{q} \in \mathbb{R}^{2^n}} \|\boldsymbol{I}_{\text{and}}\|_1 + \|\boldsymbol{I}_{\text{or}}\|_1 \quad \text{s.t.} \quad \forall S \subseteq N, |q_S| < \tau \quad (6)$$

where $\| \cdot \|_1$ represents L-1 norm function, $\boldsymbol{p} = [p_{S_1}, p_{S_2}, ..., p_{S_{2^n}}]^\top$ denotes the bias terms for all masked boards. $\boldsymbol{q} = [q_{S_1}, q_{S_2}, ..., q_{S_{2^n}}]^\top$. $\boldsymbol{I}_{\text{and}} = [I_{\text{and}}(S_1), I_{\text{and}}(S_2), ..., I_{\text{and}}(S_{2^n})]^\top$ and $\boldsymbol{I}_{\text{or}} = [I_{\text{or}}(S_1), I_{\text{or}}(S_2), ..., I_{\text{or}}(S_{2^n})]^\top$ denote AND interactions and OR interactions, respectively. AND interactions $\{I_{\text{and}}(S)\}_{S \subseteq N}$ are computed by setting $v_{\text{and}}(\boldsymbol{x}_T) = \frac{1}{2} \cdot [v(\boldsymbol{x}_T) + q_T] + p_T$, and OR interactions $\{I_{\text{or}}(S)\}_{S \subseteq N}$ are computed by setting $v_{\text{or}}(\boldsymbol{x}_T) = \frac{1}{2} \cdot [v(\boldsymbol{x}_T) + q_T] - p_T$ in Equation (4).

● **Challenge 2. Verifying that the inference logic of the value network can be explained as sparse interaction primitives.** Although Ren et al. (2023b) have proved that a well-trained DNN usually just encodes a small number of AND interactions between input variables for inference under some common conditions[3], it is still a challenge to strictly examine whether the value network fully satisfies these conditions. Although according to Theorem 2, the above OR interaction can be considered as a specific AND interaction, in real applications, we still need to verify the sparsity of interactions encoded by the value network for the Go game.

Therefore, we experimentally examine the sparsity of interactions on the KataGo (Wu, 2019), which is a free open-source neural network for the game of Go and has defeated top-level human players. We extract interactions encoded by the value network of the KataGo. Specifically, we use KataGo to generate a board state by letting KataGo take turns to play the moves of black stones and those of white stones. Let there be $m$ stones on the board. Considering the exponentially large cost of computing interactions, we just select and explain $n = 10$ stones ($n \leq m$), including $\frac{n}{2}$ white stones and $\frac{n}{2}$ black stones, and limit our attention to interactions between these $n$ stones. All other stones on the board can be considered as constant background, whose interactions are not computed. Figure 2 shows the strength $|I(S)|$ of effects of different AND interactions and OR interactions in a descending order. It shows that only a few interactions have salient effects, 80%-85% interactions have negligible effects. It verifies that interactions encoded by the value network are sparse.

● **Challenge 3. How to ensure that the inference logic of the value network can be explained as simple interaction primitives?** We find a problem that most interaction primitives extracted

---

[5]Please see Appendix F for the proof.

[6]Please see Appendix I.2 for more details about setting the small threshold $\tau$.

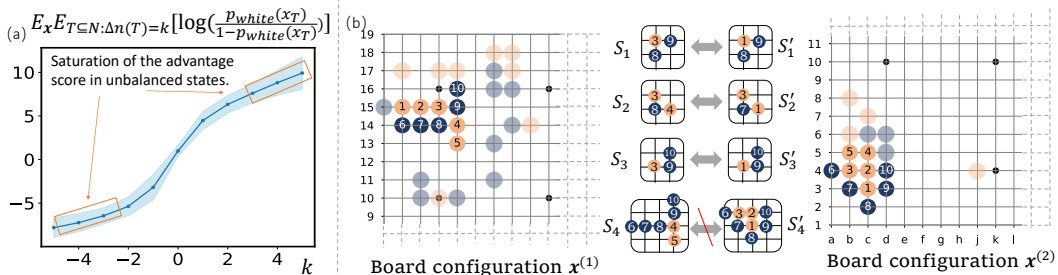

Figure 3: (a) The average advantage score $A_k = \mathbb{E}_{\boldsymbol{x}}\mathbb{E}_{T\subseteq N:\Delta n(T)=k}\log\left(\frac{p_{\text{white}}(\boldsymbol{x}_T)}{1-p_{\text{white}}(\boldsymbol{x}_T)}\right)$ over all masked states $\boldsymbol{x}_T$ with the same unbalance level $k$. The average advantage score is saturated when $|k|$ is large. (b) Compared to high-order interactions $S_4$, low-order interactions $S_1, S_2, S_3$ extracted from the board state $\boldsymbol{x}^{(1)}$ are usually easier to be transferred to another board state $\boldsymbol{x}^{(2)}$.

from the KataGo are high-order interaction primitives (see Figure 4). The high order of interaction primitives significantly boosts the difficulty of extracting common shape patterns widely used in different games. It is because high-order interactions are usually considered to be "special shapes" in a specific game, instead of simple (low-order) shape patterns frequently used in different games. For example, as Figure 3 (b) shows, 3-order interaction primitives $S_1, S_2, S_3$ extracted from the board state $\boldsymbol{x}^{(1)}$ can be transferred to another board state $\boldsymbol{x}^{(2)}$. However, the 8-order interaction primitive $S_4$ extracted from $\boldsymbol{x}^{(1)}$ cannot be transferred to another board state $\boldsymbol{x}^{(2)}$.

The reason for the emergence of high-order interactions is that most training samples for the value network are usually biased to states with similar numbers of white stones and black stones, because in real games, the board always contains similar numbers of white stones and black stones. Such a bias leads to the following saturation problem, which makes most interaction primitives be high-order primitives[7]. We use $\Delta n(T) = n_{\text{white}}(T) - n_{\text{black}}(T) \in \{-n/2, -n/2+1, ..., n/2\}$ to measure the unbalance level of the masked state $\boldsymbol{x}_T$, where $n_{\text{white}}(T)$ and $n_{\text{black}}(T)$ denote the number of white stones and that of black stones on $\boldsymbol{x}_T$, respectively. As Figure 3 (a) shows, we compute the average advantage score over all masked states $\boldsymbol{x}_T$ with the same unbalance level $k \in \{-n/2, -n/2+1, ..., n/2\}$, $A_k = \mathbb{E}_{\boldsymbol{x}}\mathbb{E}_{T\subseteq N:\Delta n(T)=k}\log\left(\frac{p_{\text{white}}(\boldsymbol{x}_T)}{1-p_{\text{white}}(\boldsymbol{x}_T)}\right)$. We find that this average advantage score $A_k = \mathbb{E}_{\boldsymbol{x}}\mathbb{E}_{T\subseteq N:\Delta n(T)=k}\log\left(\frac{p_{\text{white}}(\boldsymbol{x}_T)}{1-p_{\text{white}}(\boldsymbol{x}_T)}\right)$ is not roughly linear with the $k$ value, but is saturated when $|k|$ is large. This is the main reason for high-order interactions[7].

In order to alleviate the above saturation problem, we revise the advantage score $v(\boldsymbol{x}_T)$ in Equation (1) to remove the value shift caused by the saturation problem, i.e., $u(\boldsymbol{x}_T) = v(\boldsymbol{x}_T) - a_k$. Given a masked state $\boldsymbol{x}_T$, we compute its unbalance level $k = \Delta n(T) = n_{\text{white}}(T) - n_{\text{black}}(T) \in \{-n/2, -n/2+1, ..., n/2\}$. $a_k$ is initialized as the average advantage score $A_k$. We extend the loss function in Equation (6) as follows to learn the parameters $\boldsymbol{a} = [a_{-\frac{n}{2}}, a_{-\frac{n}{2}+1}, ..., a_{\frac{n}{2}}]^\top \in \mathbb{R}^{n+1}$.

$$\min_{\boldsymbol{p},\boldsymbol{q}\in\mathbb{R}^{2^n},\boldsymbol{a}\in\mathbb{R}^{n+1}} \|\boldsymbol{I}_{\text{and}}\|_1 + \|\boldsymbol{I}_{\text{or}}\|_1 \quad \text{s.t.} \quad \forall S \subseteq N, |q_S| < \tau \tag{7}$$

We learn parameters $\boldsymbol{p}$, $\boldsymbol{q}$, and $\boldsymbol{a}$ to obtain the sparse decomposition of AND interactions $\boldsymbol{I}_{\text{and}} = [I_{\text{and}}(S_1), I_{\text{and}}(S_2), ..., I_{\text{and}}(S_{2^n})]^\top$ and OR interactions $\boldsymbol{I}_{\text{or}} = [I_{\text{or}}(S_1), I_{\text{or}}(S_2), ..., I_{\text{or}}(S_{2^n})]^\top$. AND interactions $\{I_{\text{and}}(S)\}_{S\subseteq N}$ are computed by setting $v_{\text{and}}(\boldsymbol{x}_T) = \frac{1}{2} \cdot [u(\boldsymbol{x}_T) + q_T] + p_T$, and OR interactions $\{I_{\text{or}}(S)\}_{S\subseteq N}$ are computed by setting $v_{\text{or}}(\boldsymbol{x}_T) = \frac{1}{2} \cdot [u(\boldsymbol{x}_T) + q_T] - p_T$ in Equation (4). The small threshold $\tau = 0.38$ is set to be the same as in Equation (6)

*Penalizing high-order interactions.* Besides, we can further add another loss to Equation 7 to penalize high-order interactions, i.e., $Loss = \|\boldsymbol{I}_{\text{and}}\|_1 + \|\boldsymbol{I}_{\text{or}}\|_1 + r \cdot \|\boldsymbol{I}_{\text{and}}^{\text{high}}\|_1 + \|\boldsymbol{I}_{\text{or}}^{\text{high}}\|_1$, where $\boldsymbol{I}_{\text{and}}^{\text{high}}$ denotes a 386-dimension vector that corresponds to 386 interactions of the 6-th-10-th orders in the vector $\boldsymbol{I}_{\text{and}}$. In this loss, we set $r = 5.0$ to boost the penalty of high-order interactions.

*Experiments.* We conduct experiments to check whether above methods can reduce the complexity (order) of the extracted interactions, compared with the original interactions extracted by methods in Equation (6). Specifically, we follow experimental settings in Challenge 2 to generate a board, and

---

[7]Please see Appendix G for the reason why the saturation problem causes high-order interactions.

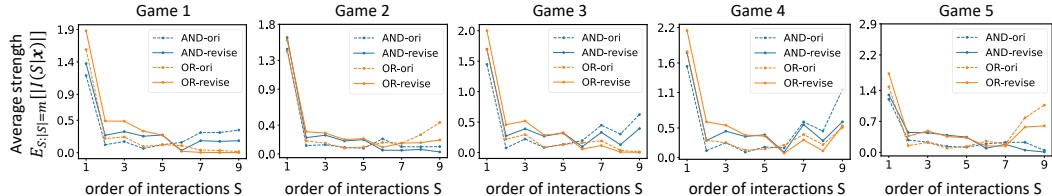

Figure 4: Average strength of effects for interactions of different orders. For different games, the revised method extracts weaker high-order interactions than the original method.

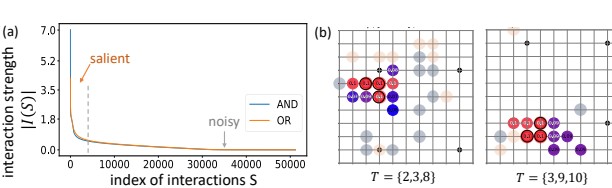

Figure 5: (a) Strength of all revised AND interactions and revised OR interactions of all 50 games in a descending order. Only a few interactions have salient effects on the output of the value network. (b) Interaction context of the coalition.

compute AND-OR interactions between selected $n$ stones. Then, we compute the average strength of AND-OR interactions of different orders, $\mathbb{E}_{S:|S|=m}[|I_{\text{and}}(S)|]$ and $\mathbb{E}_{S:|S|=m}[|I_{\text{or}}(S)|]$, respectively. Figure 4 shows the average strength of interaction effects. For both AND interactions and OR interactions, we observe that the revised method generates much weaker high-order interactions than the original method in Equation (6). This verifies the effectiveness of the revised method to reduce the complexity of the extracted interactions.

*Sparsity of interactions extracted by the revised method.* We follow experimental settings in Challenge 2 to generate 50 game states, and visualize the strength of all AND interactions and all OR interactions of all these 50 game states in a descending order. Figure 5 shows that only a few interactions have salient effects, more than 90% interactions have small effects, which verifies the sparsity of interactions extracted by the revised method.

*Still satisfying the universal matching property in Theorem 3.* Theoretically, the AND-OR interactions extracted by our revised method can still satisfy the universal matching property in Theorem 3. Furthermore, given a board state $\boldsymbol{x}$, we conduct experiments to examine whether we can use the extracted AND-OR interactions to approximate the network outputs $v(\boldsymbol{x}_T)$ on all different randomly masked board states $\{\boldsymbol{x}_T\}_{T\subseteq N}$. To this end, for each arbitrarily masked board states $\boldsymbol{x}_T$, we measure the approximation error $\Delta v_T = |v^{\text{real}}(\boldsymbol{x}_T) - v^{\text{approx}}(\boldsymbol{x}_T)|$ of using AND-OR interactions to mimic the real network output $v^{\text{real}}(\boldsymbol{x}_T)$, where $v^{\text{approx}}(\boldsymbol{x}_T) = v(\boldsymbol{x}_\emptyset) + \sum_{S\subseteq T, S\neq\emptyset} I_{\text{and}}(S) + \sum_{S\cap T\neq\emptyset, S\neq\emptyset} I_{\text{or}}(S)$ represents the score approximated by AND-OR interactions according to Theorem 3. In Figure 6, the solid curve shows the real network outputs on all $2^n$ randomly masked board states when we sort all $2^n$ network outputs in an ascending order. The shade area shows the smoothed approximation error, which is computed by averaging approximation errors of neighboring 50 masked board states. Figure 6 shows that the approximated outputs $v^{\text{approx}}(\boldsymbol{x}_T)$ can well match with the real outputs $v^{\text{real}}(\boldsymbol{x}_T)$ over different randomly masked states, which indicates that the output of the value network can be explained as AND-OR interactions.

### 3.3 DISCOVERING NOVEL SHAPES FROM THE VALUE NETWORK

In the above section, we have extracted sparse and simple interaction primitives from the value network. In this section, we aim to discover novel shapes from these interaction primitives, and use the discovered novel shapes to teach people about the game of Go.

We have examined the sparsity of interaction primitives in experiments. We can usually extract about 100–250 interaction primitives to explain the output score of a single board state. However, the number of primitives is still too large to teach people, and we need a more efficient way to discover novel shapes encoded by the value network. Therefore, we visualize all interaction primitives, and then identify some specific combinations of stones that frequently appear in different interaction primitives. We refer to these combinations as "*common coalitions.*" For example, given a board state

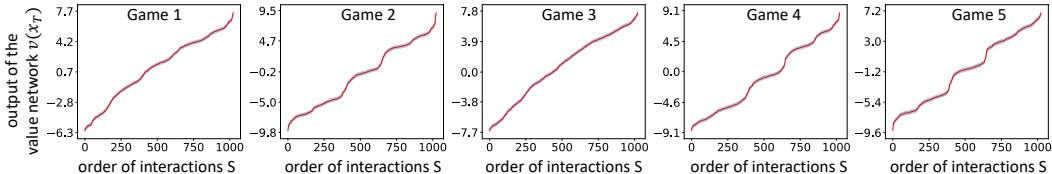

Figure 6: Outputs of the value network on all $2^n$ masked board states $v(\boldsymbol{x}_T)$ (the red full line), which are arranged in ascending order. The height of the blue shade represents the smoothed approximation error, which is computed by averaging the approximation errors $\Delta v_T = |v^{\text{real}}(\boldsymbol{x}_T) - v^{\text{approx}}(\boldsymbol{x}_T)|$ of neighboring 50 masked states.

$\boldsymbol{x}$ with $n$ stones, indexed by $N = \{1, 2, ..., n\}$ in Figure 1, we can extract some salient interactions from the board state $\boldsymbol{x}$, such as $S_1 = \{1, 3, 4, 5, 8\}, S_2 = \{1, 3, 8\}, S_3 = \{1, 3, 8, 9\}$, *etc*. The coalition $T = S_1 \cap S_2 \cap S_3 = \{1, 3, 8\}$ participates in different interactions $T \subseteq S_1, S_2, S_3$. We can consider this coalition $T$ as a *classical shape pattern* encoded by the value network.

Therefore, we further compute the attribution $\varphi(T)$ of each coalition $T$ to the advantage score $v(\boldsymbol{x})$ estimated by the value network. In this way, a positive attribution $\varphi(T) > 0$ means that the shape pattern of the coalition $T$ tends to enhance the advantage of white stones. In comparison, a negative attribution $\varphi(T) < 0$ means that the shape pattern of the coalition $T$ tends to decrease the advantage score. $\varphi(T) \approx 0$ means that although the coalition $T$ is well modeled by the value network, the coalition $T$ has contradictory effects when it appears in different interactions, thereby not making a significant effect on the advantage score.

There are a lot of attribution methods (Lundberg & Lee, 2017; Selvaraju et al., 2017; Zhou et al., 2016; Zintgraf et al., 2017) to estimate the attribution/importance score of different input variables of an AI model, *e.g.*, estimating the attributions of different image patches to the image-classification score, or the attributions of different tokens in natural language processing. However, there is no a widely accepted method to estimate the attribution of a coalition of input variables, because most attribution methods cannot generate self-consistent attribution values[8]. Therefore, we apply the method (Xinhao Zheng, 2023) to define the attribution of a coalition $T$. This method extends the theory of the Shapley value and well explains the above inconsistency problem. Specifically, the attribution score $\varphi(T)$ of the coalition $T$ is formulated as the weighted sum of effects of AND-OR interactions, as follows.

$$\varphi(T) = \sum_{S \supseteq T} \frac{|T|}{|S|} [I_{\text{and}}(S) + I_{\text{or}}(S)] \tag{8}$$

$$\varphi(T) - \sum_{i \in T} \phi(i) = \sum_{S \subseteq N, S \cap T \neq \emptyset, S \cap T \neq T} \frac{|S \cap T|}{|S|} [I_{\text{and}}(S) + I_{\text{or}}(S)] \tag{9}$$

Let there be some AND interactions and OR interactions containing the coalition $T$. Then, Equation (8) shows that for each interaction $S \supseteq T$ containing the coalition $T$, we must allocate a ratio $\frac{|T|}{|S|}$ of its interaction effect as a numerical component of $\varphi(T)$. In addition, Appendix H shows a list of theorems and properties of the attribution of the coalition defined in Equation (8), which theoretically guarantee the faithfulness of the attribution metric $\varphi(T)$. For example, Equation (9) explains the difference $\varphi(T) - \sum_{i \in T} \phi(i)$ between the coalition's attribution $\varphi(T)$ and the sum of Shapley values $\phi(i)$ for all input variables $i$ in $T$. The difference comes from those interactions that only contain partial variables in $T$, not all variables in $T$. Please see Appendix H for more theorems.

*Experiments.* Given a board state, we extract interaction primitives encoded by the value network, *i.e.*, $\{S : |I(S)| > \xi\}$, where $\xi = 0.15 \cdot \max_S |I(S)|$. Then, we manually annotate 50 coalitions based on the guidance from professional human Go players. Figure 7 visualizes sixteen coalitions selected from four game states. Figure 5 (b) shows the interaction context of the coalition. Please see Appendix I.3 for more details about the computation of the attribution of the interaction context.

---

[8]We use the following example to introduce the inconsistency problem. We can simply consider a coalition $T$ (*e.g.*, $T = \{1, 2, 3\}$) of input variables as a singleton variable $[T]$, then we have a total of $n - 2$ input variables in $N' = \{[T], 4, 5, ..., n\}$. Let $\varphi([T])$ denote the attribution of $[T]$ computed on the new partition $N'$ of the $n - 2$ variables. Alternatively, we can also consider $x_1, x_2, x_3$ as three individual variables, and compute their attributions $\varphi(1), \varphi(2), \varphi(3)$ given the original partition of input variables $N = \{1, 2, ..., n\}$. However, for most attribution methods, $\varphi([T]) \neq \varphi(1) + \varphi(2) + \varphi(3)$. This is the inconsistency problem of attributions.

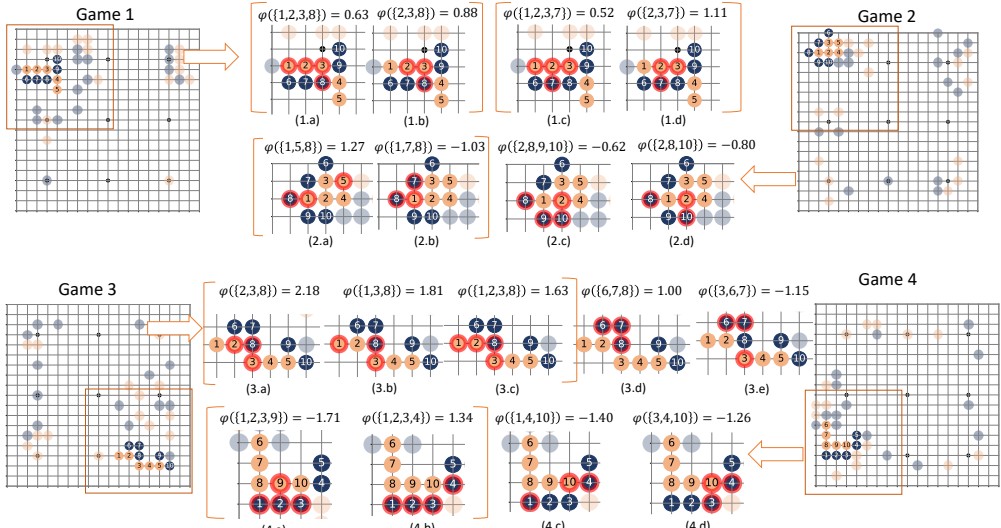

Figure 7: Estimated attributions of different coalitions (shape patterns). Stones in the coalition are high-lighted by red circles.

### 3.4 HUMAN PLAYERS' INTERPRETATION OF THE CLASSIC SHAPES/COALITIONS

In order to interpret shape patterns (coalitions) encoded by the value network, we collaborate with the professional human Go player[9]. Based on Figure 7, they find both shape patterns that fit common understandings of human players and shape patterns that conflict with human understandings.

**Cases that fit human understandings.** For the Game 1 in Figure 7 (1.a - 1.d), $\varphi(\{1, 2, 3, 8\}) < \varphi(\{2, 3, 8\})$ and $\varphi(\{1, 2, 3, 7\}) < \varphi(\{2, 3, 7\})$. It means that when the white stone $x_1$ participates in the combination of white stones $x_2, x_3$, the advantage of white stones become lower, *i.e.*, the stone $x_1$ is a low-value move. Go players consider that the effect of the combination of white stones $x_1, x_2, x_3$ is low. For the Game 2 in Figure 7 (2.a, 2.b), $\varphi(\{1, 5, 8\}) > \varphi(\{1, 2, 8\})$ means that the value network considers that the white stone $x_5$ has higher value than $x_2$. Go players consider that in this game state, the white stone $x_5$ protects the white stones $x_1, x_2, x_3, x_4$, and the white stones $x_1, x_3$ attack the black stones $x_6, x_7$, but the white stone $x_2$ has much less value than other stones. For the Game 3 in Figure 7 (3.a - 3.c), $\varphi(S_1 = \{1, 3, 8\}) > \varphi(S_3 = \{1, 2, 3, 8\})$ and $\varphi(S_2 = \{2, 3, 8\}) > \varphi(S_3 = \{1, 2, 3, 8\})$, subject to $S_3 = S_1 \cup S_2$. Go players consider that the existence of the local shape $S_1 = \{1, 3, 8\}$ makes the move of the stone $x_2$ have a low value, *i.e.*, given the context $S_1$, the stone $x_2$ wastes a move, thereby losing some advantages. Figure 7 shows some strange shape patterns that go beyond the understandings of human Go players.

**Cases that conflict with human understandings.** For Game 3 in Figure 7 (3.d, 3.e), $\varphi(\{6, 7, 8\}) = 1.00$ and $\varphi(\{3, 6, 7\}) = -1.15$, Go players are confused that the coalition $\{6, 7, 8\}$ is advantageous for white stones, and the coalition $\{3, 6, 7\}$ is advantageous for black stones. For Game 4 in Figure 7 (4.a, 4.b), $\varphi(\{1, 2, 3, 4\}) = 1.34$ and $\varphi(\{1, 2, 3, 9\}) = -1.71$. It means that the coalition $\{1, 2, 3, 4\}$ is advantageous for white stones, and the coalition $\{1, 2, 3, 9\}$ is advantageous for black stones.

## 4 CONCLUSION

In this paper, we extract sparse interactions between stones memorized by the value network for the game of Go. We regard common coalitions shared by different interactions as shape patterns, and estimate attribution values of these common coalitions. Then, we examine the fitness and conflicts between the automatically extracted shape patterns and conventional human understanding of the game of Go, so as to help human players learn novel shapes from the value network. We collaborate with professional human Go players to provide deep insights into shape patterns that are automatically extracted from the value network.

---

[9]During the review phase, the Go players are anonymous, because they are also authors.

## Ethic Statement

This paper aims to extract sparse and simple interaction primitives between stones encoded by the value network for the game of Go, thereby teaching people to learn from the value network. Previous methods usually extract AND-OR interactions to represent the primitives encoded by the AI model. However, we discover that although AND-OR interactions have some good mathematical properties, the interaction primitives (shape patterns) extracted by this method are usually extremely complex, *i.e.*, the shape patterns usually contain many stones. Such complexity of the extracted shape patterns makes it difficult for people to learn from the value network. Thus, we propose a method to extract sparse and simple interactions encoded by the value network. There are no ethic issues with this paper.

## Reproducibility Statement

We have provided proofs for the theoretical results of this study in Appendix A, B, C, D, E, F, G, H. We have also provided experimental details in Appendix I and more experimental results in Appendix J. Furthermore, we will release the code when the paper is accepted.

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

## A  PROPERTIES FOR THE HARSANYI DIVIDEND

In this paper, we follow Ren et al. (2023a) to use the Harsanyi dividend (or Harsanyi interaction) to measure the numerical effect $I(S)$ of the interaction primitive $S$. Ren et al. (2023a) have proved that the Harsanyi dividend satisfied the following properties, including the *efficiency, linearity, dummy, symmetry, anonymity, recursive, interaction distribution properties.*

(1) Efficiency property: The inference score of a well-trained model $v(\boldsymbol{x})$ can be disentangled into the numerical effects of different interaction primitives $I(S), S \subseteq N$, *i.e.*, $v(\boldsymbol{x}) = \sum_{S \subseteq N} I(S)$.

(2) Linearity property: If the inference score of the model $w$ is computed as the sum of the inference score of the model $u$ and the inference score of the model $v$, *i.e.*, $\forall S \subseteq N, w(\boldsymbol{x}_S) = u(\boldsymbol{x}_S) + v(\boldsymbol{x}_S)$, then the interactive effect of $S$ on the model $w$ can be computed as the sum of the interaction effect of $S$ on the model $u$ and that on the model $v$, *i.e.*, $\forall S \subseteq N, I_w(S) = I_u(S) + I_v(S)$.

(3) Dummy property: If the input variable $i$ is a dummy variable, *i.e.*, $\forall S \subseteq N \setminus \{i\}, v(\boldsymbol{x}_{S \cup \{i\}}) = v(\boldsymbol{x}_S) + v(\boldsymbol{x}_{\{i\}})$, then the input variable $i$ has no interaction with other input variables, *i.e.*, $\forall \emptyset \neq S \subseteq N \setminus \{i\}, I(S \cup \{i\}) = 0$.

(4) Symmetry property: If the input variable $i \in N$ and the input variable $j \in N$ cooperate with other input variables in $S \subseteq N \setminus \{i, j\}$ in the same way, *i.e.*, $\forall S \subseteq N \setminus \{i, j\}, v(\boldsymbol{x}_{S \cup \{i\}}) = v(\boldsymbol{x}_{S \cup \{j\}})$, then the input variable $i$ and the input variable $j$ have the same interactive effect, *i.e.*, $\forall S \subseteq N \setminus \{i, j\}, I(S \cup \{i\}) = I(S \cup \{j\})$.

(5) Anonymity property: If a random permutation $\pi$ is added to $N$, then $\forall S \subseteq N, I_v(S) = I_{\pi v}(\pi S)$ is always guaranteed, where the new set of input variables $\pi S$ is defined as $\pi S = \{\pi(i), i \in S\}$, the new model $\pi v$ is defined as $(\pi v)(\boldsymbol{x}_{\pi S}) = v(\boldsymbol{x}_S)$. This suggests that permutation does not change the interactive effects.

(6) Recursive property: The interactive effects can be calculated in a recursive manner. For $\forall i \in N, S \subseteq N \setminus \{i\}$, the interactive effect of $S \cup \{i\}$ can be computed as the difference between the interactive effect of $S$ with the presence of the variable $i$ and the interactive effect of $S$ with the absence of the variable $i$. *I.e.*, $\forall i \in N, S \subseteq N \setminus \{i\}, I(S \cup \{i\}) = I(S|i \text{ is consistently present}) - I(S)$, where $I(S|i \text{ is consistently present}) = \sum_{L \subseteq S}(-1)^{|S|-|L|} v(\boldsymbol{x}_{L \cup \{i\}})$.

(7) Interaction distribution property: This property describes how an interaction function Sundararajan et al. (2020b) distributes interactions. An interaction function $v_T$ parameterized by a context $T$ is defined as follows. $\forall S \subseteq N$, if $T \subseteq S$, then $v_T(\boldsymbol{x}_S) = c$; if not, $v_T(\boldsymbol{x}_S) = 0$. Then, the interactive effects for an interaction function $v_T$ can be computed as, $I(T) = c$, and $\forall S \neq T, I(S) = 0$.

## B  COMMON CONDITIONS FOR THE SPARSITY OF INTERACTIONS ENCODED BY A DNN

Ren et al. (2023b) presented the sufficient conditions for the sparsity of interaction primitives encoded by the DNN, *i.e.*, (1) the DNN does not encode interaction primitives of extremely high order, *i.e.*, the DNN does not encode too complex interaction primitives, such as encoding complex interactions between over 70 stones; (2) When the input samples are partially occluded or masked, the output of the DNN should monotonically decrease as the number of masked input variables increases; (3) The inference score of the masked input sample should not be too low, and the inference score of the normal input sample should not be too high.

## C  PROVING THAT THE OR INTERACTIONS CAN BE CONSIDERED AS A SPECIFIC AND INTERACTION

The effect $I_{\text{or}}(S)$ of an OR interaction $S$ is defined as follows.

$$I_{\text{or}}(S) = -\sum_{T \subseteq S}(-1)^{|S|-|T|} v(\boldsymbol{x}_{N \setminus T}), \quad S \neq \emptyset \tag{10}$$

Here, $\boldsymbol{x}_{N \setminus T}$ denotes the masked board state where stones in the set $N \setminus T$ are placed on the board, and stones in the set $T$ are removed. We reconsider the definition of the masked board state $\boldsymbol{x}$ as the

definition of $\boldsymbol{x}'$. In comparison, $\boldsymbol{x}'_T$ denotes the masked board state where stones in the set $T$ are removed (based on the definition of $\boldsymbol{x}_T$, stones in the set $T$ are placed on the board), and stones in the set $N \setminus T$ are placed on the board (based on $\boldsymbol{x}_T$, stones in the set $N \setminus T$ are removed).

In this way, $\boldsymbol{x}_{N \setminus T}$ denotes the same board state as $\boldsymbol{x}'_T$. The effect $I_{\text{or}}(S|\boldsymbol{x})$ of an OR interaction based on the definition of $\boldsymbol{x}$ can be reformulated as the effect $I'_{\text{and}}(S|\boldsymbol{x}')$ of an AND interaction based on the definition of $\boldsymbol{x}'$ as follows.

$$
\begin{aligned}
I_{\text{or}}(S|\boldsymbol{x}) &= -\sum\nolimits_{T \subseteq S} (-1)^{|S|-|T|} v(\boldsymbol{x}_{N \setminus T}), \quad S \neq \emptyset \\
&= -\sum\nolimits_{T \subseteq S} (-1)^{|S|-|T|} v(\boldsymbol{x}'_T), \quad S \neq \emptyset \\
&= -I'_{\text{and}}(S|\boldsymbol{x}'), \quad S \neq \emptyset
\end{aligned}
\tag{11}
$$

Therefore, we consider the OR interaction as a specific AND interaction.

## D  PROVING THAT THE MODEL OUTPUT CAN BE REPRESENTED AS OR INTERACTIONS

According to Appendix C, we reconsider the definition of the masked board state $\boldsymbol{x}_T$ as $\boldsymbol{x}'_T$. $\boldsymbol{x}_T$ denotes the masked board state where stones in the set $T$ are placed on the board, and stones in the set $N \setminus T$ are removed. In comparison, $\boldsymbol{x}'_T$ denotes the masked board state where stones in the set $T$ are removed, and stones in the set $N \setminus T$ are placed on the board.

In this way, the effect of an OR interaction based on the definition of $\boldsymbol{x}$ can be represented as the effect $I'_{\text{and}}(S|\boldsymbol{x}')$ of an AND interaction based on the definition of $\boldsymbol{x}'$.

$$
\begin{aligned}
I_{\text{or}}(S|\boldsymbol{x}) &= w_S^{\text{or}} \cdot [-\prod\nolimits_{i \in S} \neg exist(x_i)] \\
&= -w_S^{\text{or}} \cdot \prod\nolimits_{i \in S} \neg exist(x_i) \\
&= -\frac{w_S^{\text{or}}}{w_S^{\text{and}}} \cdot I'_{\text{and}}(S|\boldsymbol{x}')
\end{aligned}
\tag{12}
$$

where the function $exist(x_i)$ represents that the stone $x_i$ is placed on the board, the function $\neg exist(x_i)$ represents that the stone $x_i$ is removed from the board.

## E  PROVING THAT THE NETWORK OUTPUT CAN BE REPRESENTED AS AND-OR INTERACTIONS

We derive that for all $2^n$ randomly masked sample $\boldsymbol{x}_T$, the output score $v(\boldsymbol{x}_T)$ of the DNN on $\boldsymbol{x}_T$ can be approximated by the sum of effects of AND-OR interactions, i.e., $v(\boldsymbol{x}_T) = v(\boldsymbol{x}_\emptyset) + \sum_{S \subseteq T, S \neq \emptyset} I_{\text{and}}(S) + \sum_{S \cap T \neq \emptyset, S \neq \emptyset} I_{\text{or}}(S)$

$$
\begin{aligned}
\sum\nolimits_{S \subseteq T} I_{\text{and}}(S) &= \sum\nolimits_{S \subseteq T} \sum\nolimits_{L \subseteq S} (-1)^{|S|-|L|} v_{\text{and}}(\boldsymbol{x}_L) \\
&= \sum\nolimits_{L \subseteq T} \sum\nolimits_{S: L \subseteq S \subseteq T} (-1)^{|S|-|L|} v_{\text{and}}(\boldsymbol{x}_L) \\
&= \underbrace{v_{\text{and}}(\boldsymbol{x}_T)}_{L=T} + \sum\nolimits_{L \subseteq T, L \neq T} v_{\text{and}}(\boldsymbol{x}_L) \cdot \underbrace{\sum\nolimits_{m=0}^{|T|-|L|} (-1)^m}_{=0} \\
&= v_{\text{and}}(\boldsymbol{x}_T)
\end{aligned}
\tag{13}
$$

$$\sum_{S\cap T\neq\emptyset,S\neq\emptyset} I_{\text{or}}(S) = -\sum_{S\cap T\neq\emptyset,S\neq\emptyset}\sum_{L\subseteq S}(-1)^{|S|-|L|}v_{\text{or}}(\boldsymbol{x}_{N\setminus L})$$

$$= -\sum_{L\subseteq N}\sum_{S:S\cap T\neq\emptyset,S\supseteq L}(-1)^{|S|-|L|}v_{\text{or}}(\boldsymbol{x}_{N\setminus L})$$

$$= -\underbrace{v_{\text{or}}(\boldsymbol{x}_\emptyset)}_{L=N} - \underbrace{v_{\text{or}}(\boldsymbol{x}_T)}_{L=N\setminus T}\cdot\underbrace{\sum_{|S_2|=1}^{|T|}C_{|T|}^{|S_2|}(-1)^{|S_2|}}_{=-1}$$

$$-\sum_{L\cap T\neq\emptyset,L\neq N}v_{\text{or}}(\boldsymbol{x}_{N\setminus L})\cdot\sum_{S_1\subseteq N\setminus T\setminus L}\underbrace{\sum_{|S_2|=|T\cap L|}^{|T|}C_{|T|-|T\cap L|}^{|S_2|-|T\cap L|}(-1)^{|S_1|+|S_2|}}_{=0}$$

$$-\sum_{L\cap T=\emptyset,L\neq N\setminus T}v_{\text{or}}(\boldsymbol{x}_{N\setminus L})\cdot\underbrace{\sum_{S_2\subsetneqq T}\sum_{S_1\subseteq N\setminus T\setminus L}(-1)^{|S_1|+|S_2|}}_{=0}$$

$$= v_{\text{or}}(\boldsymbol{x}_T) - v_{\text{or}}(\boldsymbol{x}_\emptyset)$$

$$(14)$$

Therefore, $v_{\text{or}}(\boldsymbol{x}_T) = \sum_{S\cap T\neq\emptyset,S\neq\emptyset} I_{\text{or}}(S) + v_{\text{or}}(\boldsymbol{x}_\emptyset)$. In this way, we can derive that the output score $v(\boldsymbol{x}_T)$ of the DNN on $\boldsymbol{x}_T$ can be approximated by the sum of effects of AND-OR interactions.

$$v(\boldsymbol{x}_T) = v_{\text{and}}(\boldsymbol{x}_T) + v_{\text{or}}(\boldsymbol{x}_T)$$
$$= \sum_{S\subseteq T} I_{\text{and}}(S) + \sum_{S\cap T\neq\emptyset,S\neq\emptyset} I_{\text{or}}(S) + v_{\text{or}}(\boldsymbol{x}_\emptyset)$$
$$= \sum_{S\subseteq T,S\neq\emptyset} I_{\text{and}}(S) + v_{\text{and}}(\boldsymbol{x}_\emptyset) + \sum_{S\cap T\neq\emptyset,S\neq\emptyset} I_{\text{or}}(S) + v_{\text{or}}(\boldsymbol{x}_\emptyset)$$
$$= \sum_{S\subseteq T,S\neq\emptyset} I_{\text{and}}(S) + \sum_{S\cap T\neq\emptyset,S\neq\emptyset} I_{\text{or}}(S) + v(\boldsymbol{x}_\emptyset)$$

$$(15)$$

## F  PROVING THAT UNAVOIDABLE NOISES IN NETWORK OUTPUT WILL ENLARGED IN INTERACTIONS

Actually, the real data inevitably contains some small noises/variations, such as texture variations and the shape deformation in object classification. Therefore, the network output $v(\boldsymbol{x}_T)$ also contains some unavoidable noises. Let $Var[v(\boldsymbol{x}_T)]$ denote the variance of the network output $v(\boldsymbol{x}_T)$, we assume that different masked input samples are independent of each other and have no correlation, then we can derive the variance of the AND interaction as follows.

$$Var[I_{\text{and}}(S)] = Var[\sum_{T\subseteq S}(-1)^{|S|-|T|}v(\boldsymbol{x}_T)]$$
$$= \sum_{T\subseteq S} Var[v(\boldsymbol{x}_T)]$$

$$(16)$$

Therefore, we prove that unavoidable noises in network output will enlarged in interactions.

## G  THE REASON WHY THE SATURATION PROBLEM CAUSES HIGH-ORDER INTERACTIONS

Let $e_k \stackrel{\text{def}}{=} \mathbb{E}_{\boldsymbol{x}}\mathbb{E}_{T\subseteq N:\Delta n=k}\log(\frac{p_{\text{white}}(\boldsymbol{x}_T)}{1-p_{\text{white}}(\boldsymbol{x}_T)})$ denote the average advantage score over all masked states $\boldsymbol{x}_T$ with the same unbalance level $k$. Let $g\in R$ and $h\in R$ denote the first derivative and second derivative of the curve of $e_k \stackrel{\text{def}}{=} \mathbb{E}_{\boldsymbol{x}}\mathbb{E}_{T\subseteq N:\Delta n=k}\log(\frac{p_{\text{white}}(\boldsymbol{x}_T)}{1-p_{\text{white}}(\boldsymbol{x}_T)})$ w.r.t. the $k$ value ($k\in\{-\frac{n}{2},-\frac{n}{2}+1,...,\frac{n}{2}\}$). Then, we can roughly consider that $e_k = e_0 + g\cdot k + \frac{h}{2}\cdot k^2$.

Let us consider an interaction $S$ between $m$ stones, including $m_{\text{white}}$ white stones and $m_{\text{black}}$ black stones. The unbalance level of the masked board state $\boldsymbol{x}_S$ is $\Delta n = m_{\text{white}} - m_{\text{black}} = k^*$. If we only use AND interactions to explain the output of the value network, then we obtain the following equation.

$$v_m \overset{\text{def}}{=} \mathbb{E}_{T \subseteq S:|T|=m}[v(\boldsymbol{x}_T)] \approx e_{k^*}$$

$$v_{m'} \overset{\text{def}}{=} \mathbb{E}_{T \subseteq S:|T|=m'}[v(\boldsymbol{x}_T)]$$
$$\approx \frac{e_{k^*-((m-m'))}}{\binom{m_{\text{white}}}{m-m'}} + \frac{e_{k^*-((m-m'-1))}}{\binom{m_{\text{white}}}{m-m'-1}} + \cdots + \frac{e_{k^*+((m-m'-1))}}{\binom{m_{\text{black}}}{m-m'-1}} + \frac{e_{k^*+((m-m'))}}{\binom{m_{\text{black}}}{m-m'}} \tag{17}$$

$$v_0 \overset{\text{def}}{=} \mathbb{E}_{T \subseteq S:|T|=0}[v(\boldsymbol{x}_T)] \approx e_0$$

Note that $v_m$, $v_{m'}$ and $v_0$ are non-linear functions. The function $v_{m'}$ can be rewritten by following Taylor series expansion at the baseline point $m' = 0$ as follows.

$$v_{m'} = \mathbb{E}_{T \subseteq S:|T|=m'}[v(\boldsymbol{x}_T)] = v_0 + g_v \cdot m' + \frac{h_v}{2} \cdot m'^2 \tag{18}$$

where $g_v \in R$ and $h_v \in R$ denote the first derivative and second derivative of the curve of $v_{m'}$ w.r.t. the $m'$ value. In this way, the effect $I(S)$ of the interaction $S$ can be reformulated as follows.

$$I(S) = \sum_{T \subseteq S} (-1)^{|S|-|T|} v(\boldsymbol{x}_T)$$
$$\approx \binom{m}{0} v_m - \binom{m}{1} \cdot v_{m-1} + \binom{m}{2} \cdot v_{m-2} - \binom{m}{3} \cdot v_{m-3} + \binom{m}{4} \cdot v_{m-4} - \dots \tag{19}$$

According to Equation (18), each component $v_{m'}$ of $I(S)$ consists of a term $\frac{h_v}{2} \cdot m'^2$. However, the term $\frac{h_v}{2} \cdot m'^2$ contained in $v_{m'}$ cannot cancel out with each other. Therefore, the interaction effect $I(S)$ will increase with the order of the primitive $S$.

## H  THEOREMS AND PROPERTIES OF THE ATTRIBUTION METHOD IN EQUATION (8).

The coalition attribution satisfies the following desirable properties.

• **Symmetry property:** If the input variable $i \in N$ and the input variable $j \in N$ cooperate with other input variables in $S \subseteq N \setminus \{i, j\}$ in the same way, i.e. $\forall S \subseteq N \setminus \{i,j\}, v(S \cup \{i\}) = v(S \cup \{j\})$, then the coalition formed by $S \cup \{i\}$ and the coalition formed by $S \cup \{j\}$ have the same attribution, i.e., $\forall S \subseteq N \setminus \{i,j\}, \varphi(S \cup \{i\}) = \varphi(S \cup \{j\})$.

• **Additivity property:** If the output score of the model $v$ can be represented as the sum of the output score of the model $v_1$ and the output score of the model $v_2$, i.e. $\forall S \subseteq N, \ v(S) = v_1(S) + v_2(S)$, then the attribution of any coalition $S$ on the model $v$ can also be represented as the sum of the attribution of $S$ on the model $v_1$ and that on the model $v_2$, i.e. $\forall S \subseteq N, \ \varphi_v(S) = \varphi_{v_1}(S) + \varphi_{v_2}(S)$.

• **Dummy property:** If a coalition $S$ is a dummy coalition, i.e. $\forall i \in S, \forall T \subseteq N \setminus \{i\}, v(T \cup \{i\}) = v(T)$, then the coalition $S$ has no attribution on the model output, i.e. $\varphi(S) = 0$.

• **Efficiency property:** For any coalition $S$, the model output can be decomposed into the attribution of the coalition $S$ and the attribution of each input variable in $N \setminus S$ and the utilities of the interactions covering partial variables in $S$, i.e., $\forall S \subseteq N, v(N) - v(\emptyset) = \varphi(S) + \sum_{i \in N \setminus S} \varphi(i) +$
$\sum_{T \subseteq N, T \cap S \neq \emptyset, T \cap S \neq S} \frac{|T \cap S|}{|T|} [I_{and}(T) + I_{or}(T)]$

And we try to use Corollary 4 and Equation (9) to explain the conflict between the Shapley value of input variables and the attribution of the coalition as follows.

**Corollary 4.** *If* $\forall T \subseteq N, T \ni i, T \not\supseteq S, I_{and}(T) = I_{or}(T) = 0$, *then* $\phi(i) = \frac{1}{|S|} \varphi(S)$

Corollary 4 shows that if a set $S$ of input variables is always memorized by the DNN as a coalition, and the DNN does not encode any interactions between a set $T$ of input variables, where $T$ only

contains partial variables in $S$, *i.e.*, $\forall T \subseteq N, T \cap S \neq S, T \cap S \neq \emptyset, I_{\text{and}}(T) = I_{\text{or}}(T) = 0$, then the attribution $\varphi(S)$ of the coalition $S$ can be fully determined by the sum of the Shapley value $\phi(i)$ of all input variables in $S$. Otherwise, if the DNN encodes interactions between a set $T$ of input variables, where $T$ contains just partial but not all variables in $S$, then Equation (9) shows the conflict between individual variables' attributions and the coalition $S$'s attribution come from interactions containing just partial but not all variables in $S$.

## I  EXPERIMENTAL DETAILS

### I.1  SETTINGS FOR THE GENERATION OF ONE BOARD CONFIGURATION

We use pre-trained networks published on https://github.com/lightvector/KataGo. We set the board size as 19*19, by letting the KataGo play games against itself, *i.e.*, letting the KataGo take turns to play the move of black stones and play the move of white stones, we can generate a board state.

### I.2  SETTINGS FOR THE EXTRACTION OF INTERACTIONS IN CHALLENGE 3.

The learning rate for the learnable vector $\boldsymbol{p}, \boldsymbol{q}$ exponentially decays from 1e-6 to 1e-7. In particular, each element $a_k$ in the vector $\boldsymbol{a}$ has different initial learning rates. Specifically, the learning rate of $a_k$ decayed from $\frac{1}{\binom{|k|}{\frac{n}{2}}} \cdot 1e-6$ to $\frac{1}{\binom{|k|}{\frac{n}{2}}} \cdot 1e-7$.

The threshold $\tau$ is a small scalar to bound unavoidable noises $\boldsymbol{q}$ in the network output, which is set to be $\tau = 0.38$ in experiments, which is set to be 0.01 time of the average strength of the top-1% most salient interaction. Specifically, we compute all AND interactions $\{I_{\text{and}}(S_1), I_{\text{and}}(S_2), ..., I_{\text{and}}(S_{2^n})\}$ by setting $v_{\text{and}}(\boldsymbol{x}_T)$ as $v(\boldsymbol{x}_T)$, and compute all OR interactions $\{I_{\text{or}}(S_1), I_{\text{or}}(S_2), ..., I_{\text{or}}(S_{2^n})\}$ by setting $v_{\text{or}}(\boldsymbol{x}_T)$ in Equation (4) as $v(\boldsymbol{x}_T)$. Then, all AND interactions $\{I_{\text{and}}(S_1), I_{\text{and}}(S_2), ..., I_{\text{and}}(S_{2^n})\}$ and all OR interactions $\{I_{\text{or}}(S_1), I_{\text{or}}(S_2), ..., I_{\text{or}}(S_{2^n})\}$ are arranged in descending order of their interaction strength.

### I.3  COMPUTING THE ATTRIBUTION OF THE INTERACTION CONTEXT.

The attribution of the stone in the interaction context can be computed as:

$$\text{attribution}(x_i) = \sum_S \frac{|I(S|x_i \in S)|}{|S|} \tag{20}$$

## J  MORE EXPERIMENTAL RESULTS

We show more shape patterns extracted from the value network for the game of Go.

For Game 1 in Figure 8 (1.a), Go players are confused about why the coalition $\{7, 8, 9\}$ is advantageous for black stones. For Game 2 in Figure 8 (2.a), Go players cannot figure out why the coalition $\{6, 7, 8\}$ is advantageous for white stones. For Game 3 in Figure 8, Go players consider that the black stones $x_6, x_7$ are caught, and the white stones are in advantage. However, the value network think that the coalition $\{2, 7, 8, 9\}$ and the coalition $\{4, 6, 8, 9\}$. Go players are confused about that. For Game 4 in Figure 8 (4.a-4.d), $\varphi(\{1, 2, 9\}) < \varphi(\{1, 2, 3, 9\})$, which means that the black stone $x_3$ is a low-value move, Go players consider that the stone $x_3$ a valuable move.

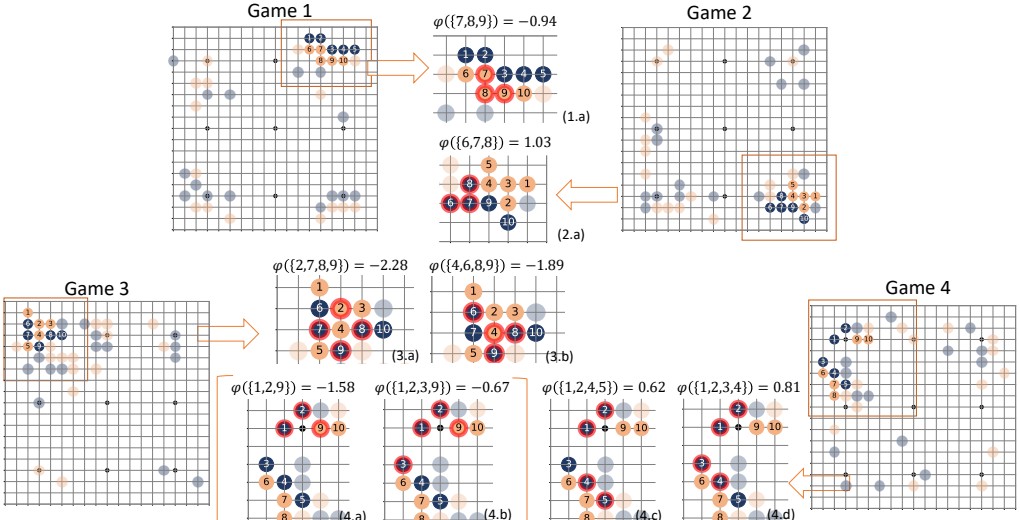

Figure 8: More experimental results for the estimated attributions of different coalitions (shape patterns). Stones in the coalition are high-lighted by red circles.

