# OpenReview forum: "Explaining How a Neural Network Play the Go Game and Let People Learn"
_ICLR.cc/2024/Conference — ICLR 2024 Conference Withdrawn Submission_

### Official Review · Reviewer_ZNeH · 2023-10-27

**Soundness:** 1 poor
**Presentation:** 1 poor
**Contribution:** 1 poor
**Rating:** 1
**Confidence:** 4

**Summary:**

This paper investigates how to interpret the neural network in the game of Go. It modeled the value of a given game state into the interactions between the stones in the game of Go. For these interactions, the authors not only discuss the original AND relationship but also extend OR relationships to it. The paper also proposes and proves several theorems showing that the output score of a random masked input x can be expressed as the sum of both AND and OR interactions. The experiments show that the interaction formulas have met the theorems through suitable modifications such as modifying the loss function. Finally, some cases of shape patterns are raised to interpret the game state and have some professional go players to see whether the patterns fit common sense or not.

**Strengths:**

The study of extracting the Go knowledge from neural networks is interesting and challenging. However, the presentation of this paper is poor, making it nearly impossible for the reader to understand. Please refer to weaknesses for more details.

**Weaknesses:**

The presentation of this paper is poor, making it nearly impossible for the reader to understand. This paper has several weaknesses listed as follows.

First, the methods are not clear. The most possible way that I can imagine is that the authors train a model to learn the weights of the intersections to fit the AlphaZero value network. The two proposed loss functions (equation 6 and equation 7) and parameters p, and q are not explained. It would be more convincing to introduce some related works listed in the paper, and clearly explain the improvement of the proposed method.

Second, the experiments are not enough to explain the Go games, and there are no baselines for comparison in the experiments. Although the authors have stated the large cost of computing interactions, conducting experiments only on 10 stones is not enough to explain the whole position. The criterion for choosing these 10 stones from the board position is also not explicitly specified.  Besides, Penalizing high-order interactions in section 3.2 might make larger patterns less primitive which contradicts the intuition of a human player to see a whole game state. Finally, the codes and data are also not available. This makes it difficult for the reader to reproduce the experiments.

Third, the results don't meet the goal of "let people learn". In section 3.4, the authors select some patterns for illustrations, but these patterns are not common (also weird) for a Go player. These patterns in Section 3.4 seem to be cherry-picked and still require Go players to explain them. It would be convincing to choose some famous patterns (e.g. knight move, bamboo) for illustrations. Overall, it is not convincing that people can learn from these extracted patterns. In addition, there are no further comparisons to other related works (Lundberg & Lee, 2017; Selvaraju et al., 2017; Zhou et al., 2016; Zintgraf et al., 2017).

Fourth, although this paper proposed several theorems and gave some strict mathematical analysis for explaining the relationships between stones, the method and experiment are not clear. For example, some proposed ideas, especially for the OR relationships are already discussed in the paper "Technical note: Defining and Quantifying AND-OR Interactions for Faithful and Concise Explanation of DNNs" in arxiv 2304.13312 but not cited. According to the statements in the introduction, "We overcome following three major challenges.", the first challenge states "... extend the original definition of interactions to further explain OR relationships …". It seems the authors claimed that it is the new ideas that they proposed. The authors should address this issue carefully.

Below are some minor comments and typos:
* Do not include citations in the abstract.
* It is better to put the introduction of Harsanyi dividend (or the Harsanyi interaction) to the related work.
* Please cite AlphaGo or AlphaZero papers when value network or policy network is referred to.
* "8-order interaction primitive S_4" should be "7-order interaction primitive S_4"
* In the references, the author list format in the paper "Towards attributions of input variables in a coalition." is wrong.
* In section 3.4, "φ({1, 5, 8}) > φ({1, 2, 8})" should be "φ({1, 5, 8}) > φ({1, 7, 8})".

**Questions:**

1. The value network of AlphaZero is ranged in [-1,1]. However, according to Figure 6, the values are not bounded in the range [-1, 1]. It could be up to 9.1 or down to -9.6. Is this value network the same as AlphaZero? If not, the author should add the definition of the value network.
2. How are the 10 stones chosen in the experiments? Why 10?
3. How many games and how much training/evaluation time are used for the experiment?
4. How are the interactions computed?

---

### Official Review · Reviewer_6nKX · 2023-10-31

**Soundness:** 1 poor
**Presentation:** 1 poor
**Contribution:** 2 fair
**Rating:** 1
**Confidence:** 3

**Summary:**

This paper seeks to extract salient board patterns of the game Go encoded using a trained value network so that human players can gain insights into gameplay and behavior. Their focus is to isolate a small set of stones in a particular shape that cause a fixed and verifiable effect on value network’s output, even in the scenario that other stones are removed from the board much like Shapley values. They identify and address 3 specific challenges. They work with professional Go players to go over some of these extracted board shapes and their effects on the value network.

**Strengths:**

1. Prior work focusses on "AND" stone interactions i.e. presence of all stones in a small set. The authors recognize that the game Go is usually more complex and they extend prior work to also include the effect of "OR" interactions on the value network.
2. Prior work has established that a well-trained DNN usually just encodes a small number of AND interactions between input variables for inference under some conditions. The authors do a small experiment on KataGo agent to examine the sparsity of interactions and discover that 85-90% of stone interactions have negligible effect.
3. They address the saturation problem. Most training samples for the value network are biased to states with equal numbers of white stones and black stones, because in real games, the board always contains similar numbers of white stones and black stones. This causes  what the authors call the "saturation" problem. They address this by revising the advantage score to remove the value shift caused by the saturation problem.
4. They work with professional Go players and analyse shapes and strategies that agree as well as conflict with human understanding so that humans can derive new insights

**Weaknesses:**

1.  The paper may not be for general purpose board game understanding, it is a method adapted solely for Go
2. The method is quite complex and paper is dense and hard to follow in a single reading. It requires multiple reading sessions to get a sense of basic flow and contributions.
3. Some of the experimental setups are unclear. Results seem to be constrained to particular settings such as only consider 10 stones or seem to be very specific board shape examples. Even if the method is sound, it is evaluated in approximate settings since stone interactions can be exponential in number.
4. I am not confident that this method is relevant beyond analyzing Go value network.

**Questions:**

1. Can the authors share more on the experiment that verifies sparsity of interactions? When only 10 stones are analyzed and rest of the stones are assumed to be noise, how confident can we be that the "noisy" stones are truly not affecting the 10 stones that we are interested in?
2. The main paper talks about two coalitions where humans and value network disagree. Can the authors share more about whose interpretation is actually the correct one in these two cases? And if humans actually were able to get new insights?

---

### Official Review · Reviewer_PcUP · 2023-11-01

**Soundness:** 1 poor
**Presentation:** 1 poor
**Contribution:** 2 fair
**Rating:** 3
**Confidence:** 3

**Summary:**

The authors present a method of identifying patterns in go based on neural network's changes in outputs when masked.

**Strengths:**

If the assumptions about linearity hold then the results may be useful for building analysis of DNNs.

**Weaknesses:**

The main result is based on assumptions of linearity and sparsity that the authors do not prove. They say that "... a well-trained DNN usually just encodes a small number of AND interactions between input variables for inference under some common conditions" and did some simple tests with KataGo and didn't do anything else to verify this assumption. The issue of DNNs being difficult to understand is in a large part due to simplifying assumptions not holding, so assuming that the system is simple makes the method highly suspect.

I also found the paper hard to follow as the abstract is very vague and the the outcome measure is never fully explained. The experiments are also very small, with only a few games/positions being discussed.

This paper seems to introduce a variation of Shapley values but only briefly mentions them, better framing of the new method relative to other methods would make the paper easier to follow and explain the contributions better.

**Questions:**

_Go game is widely considered as much more complex than most other games_ How do your citations support this claim?
Which KataGo was used?
How does this method compare to probing? McGrath, Thomas, et al. "Acquisition of chess knowledge in alphazero." Proceedings of the National Academy of Sciences 119.47 (2022): e2206625119.